# Thin-Film Flexible Wireless Pressure Sensor for Continuous Pressure Monitoring in Medical Applications

**DOI:** 10.3390/s20226653

**Published:** 2020-11-20

**Authors:** Muhammad Farooq, Talha Iqbal, Patricia Vazquez, Nazar Farid, Sudhin Thampi, William Wijns, Atif Shahzad

**Affiliations:** 1Smart Sensors Lab, School of Medicine, National University of Ireland Galway, H91 TK33 Galway, Ireland; t.iqbal1@nuigalway.ie (T.I.); patricia.vazquez@nuigalway.ie (P.V.); william.wyns@nuigalway.ie (W.W.); atif.shahzad@nuigalway.ie (A.S.); 2School of Physics, National University of Ireland Galway, H91 TK33 Galway, Ireland; nazar.farid@nuigalway.ie; 3Lambe Institute of Translational Research, National University of Ireland Galway, H91 TK33 Galway, Ireland; sudhin.thampi@nuigalway.ie

**Keywords:** pressure sensors, compression therapy, thin-film sensors, wireless sensors, medical pressure monitoring, capacitive sensors, flexible sensors, LC sensor, wound monitoring

## Abstract

Physiological pressure measurement is one of the most common applications of sensors in healthcare. Particularly, continuous pressure monitoring provides key information for early diagnosis, patient-specific treatment, and preventive healthcare. This paper presents a thin-film flexible wireless pressure sensor for continuous pressure measurement in a wide range of medical applications but mainly focused on interface pressure monitoring during compression therapy to treat venous insufficiency. The sensor is based on a pressure-dependent capacitor (C) and printed inductive coil (L) that form an inductor-capacitor (LC) resonant circuit. A matched reader coil provides an excellent coupling at the fundamental resonance frequency of the sensor. Considering varying requirements of venous ulceration, two versions of the sensor, with different sizes, were finalized after design parameter optimization and fabricated using a cost-effective and simple etching method. A test setup consisting of a glass pressure chamber and a vacuum pump was developed to test and characterize the response of the sensors. Both sensors were tested for a narrow range (0–100 mmHg) and a wide range (0–300 mmHg) to cover most of the physiological pressure measurement applications. Both sensors showed good linearity with high sensitivity in the lower pressure range <100 mmHg, providing a wireless monitoring platform for compression therapy in venous ulceration.

## 1. Introduction

Physiological pressure, including intraocular, intracranial, and cardiovascular pressure, is a key parameter for the assessment of human health and provides opportunities for early diagnosis, personalized therapy, and preventive healthcare [1]. Pressure monitoring has been used in diagnosing lower limb problems, muscle rehabilitation, and wound monitoring [1,2,3,4]. A common medical application of non-invasive pressure sensing is the monitoring of compression therapy to treat venous leg ulcers. Venous insufficiency occurs when blood is unable to return to the heart and accumulates in the lower limbs. Chronic venous insufficiency (CVI) may cause swelling, pain, edema, and ulcerations [5,6]. The most effective treatment for CVI is compression therapy, in which a compression bandage is used to apply gradual pressure between the ankle and knee to improve the circulation of blood in the lower limb [7,8]. The typical pressure range for compression therapy is between 10 and 50 mmHg, where the bandage pressure has direct impact on the healing of ulcer [4,9]. To improve the healing process of venous ulcers, continuous monitoring of applied pressure is essential and has become the focus of current research and commercial solutions. Clinical evidence suggests that compression therapy becomes more effective with a feedback sensing system. This feedback system is achieved by using a pressure sensor. Existing solutions that are commonly used in clinical practice are accurate and robust, but they are mostly tethered, rigid, bulky, and require an additional power supply [2,10,11].

The need for wireless, small scale, lightweight, and mobile sensing solutions has led current research to focus on miniaturized thin-film and microelectromechanical (MEMS) pressure sensing devices [12,13]. Current pressure monitoring technologies are generally based on either pneumatic, fluid-filled, piezoelectric, resistive, or capacitive working principles [14,15]. In a pneumatic pressure sensing system, the force of compression bandage is transferred to air pressure and later this air pressure is converted into an electrical signal for further processing [16]. Pneumatic sensors are cheap, flexible, and thin but they are not suitable for dynamic pressure applications and are prone to temperature drift and hysteresis [7,15]. Fluid-filled pressure sensors are similar to pneumatic pressure sensors, where water or oil is used instead of air [17]. The main drawbacks of a fluid-filled sensing system are the air bubbles in the fluid, leakage risk, and bulkiness [16]. In the piezoelectric sensing technology, when pressure is applied on a piezoelectric material, it gets polarized and generates a voltage differential across the device. The piezoelectric effect is proportional to the applied pressure on the device. Thin-film piezoelectric pressure sensors are used for arterial pulse monitoring, respiratory rate, and integrated with a catheter for intravascular pressure measurements, and biomedical implants [18,19,20]. Piezoelectric pressure sensors are self-powered, low cost, and good for dynamic pressure applications, but they are not suitable for static pressure measurements due to current leakages [21,22]. In resistive pressure sensing technology, the contact area between the active thin-film resistive layer and the electrodes changes with the applied pressure so the effective resistance of the sensor changes [23,24]. Resistive pressure sensors are easier to fabricate, faster in response, and less expensive than piezoelectric pressure sensors; however, an active power source with additional adapting circuitry is required to enable pressure sensing and they are very sensitive to temperature [25]. In capacitive pressure sensing technology, the distance between capacitor electrodes is a function of the applied pressure. A capacitive sensor can be either an active sensing device where the applied pressure can be measured by the changes in capacitance or more often a passive wireless sensing device by combining it with an inductor coil [8]. The combination of the sensing capacitor and inductor coil makes an inductor-capacitor (LC) resonant tank circuit, which makes it suitable for wireless sensing via inductive coupling with an external antenna. The pressure is measured from relative changes in the resonance frequency of the LC resonant tank [26,27,28]. Due to this wireless communication between sensor and reader coil, capacitive pressure sensors are more practical for wearable and implantable applications as compared to resistive and piezoelectric sensing technologies that demand wired connection to communicate. Capacitive pressure sensing technology is generally used in MEMS and thin-film pressure sensors. MEMS-based sensors are accurate, miniaturized, wirelessly powered, and are widely used in wearable and implantable applications [29]. However, MEMS sensors are generally rigid and have a complex fabrication process that requires specialized equipment. On the other hand, thin-film based capacitive pressure sensors are flexible, less expensive, and simple to fabricate [30].

In past decades, many commercial solutions have been developed for pressure monitoring during compression therapy with growing research focused on lightweight, flexible, and wireless sensing systems. PicoPress (Microlab Electronica, Ponte S. Nicolo, Italy), air-pack type analyzer (AMI Techno, Tokyo, Japan), Kikuhime pneumatic transducer (Advancis Medical, Nottinghamshire, UK), Medical stocking tester (MST, Salzmann AG (SAG), St. Gallen, Switzerland), SIGaT tester (Ganzoni-Sigvaris, St. Gallen, Switzerland), and Oxford pressure monitor MK II (Talley Ltd., Romsey, UK) are available pneumatic sensor-based solutions to monitor the pressure during compression therapy [31,32,33,34]. A comparative study has confirmed that PicoPress and Kikuhime are more accurate compared to SIGaT [31]. PicoPress, Kikuhime, MST, and SIGaT are the most common medical devices focused on clinical applications, with relatively higher costs compared to stand-alone sensors [4,8]. Because of the pneumatic sensing principle, these systems are not appropriate for continuous dynamic pressure measurements [9,34].

On the other hand, Quantum tunneling composite (QTC, Peratech, Richmond, UK), ThruMode Force Sensing Resistor (FSR, Sensitronics, Bow, WA, USA), Interlink FSR (Interlink Electronics Inc., Camarillo, CA, USA), F-Scan (Tekscan, Inc., Boston, MA, USA) and Tactilus (Sensor Products Inc., Madison, NJ, USA) are commercially available piezoresistive pressure sensors being widely used to measure interface pressure during compression therapy [35]. Although these sensors are low-cost, thin, and flexible, they require a wired connection and additional electronics to work which makes the system bulky and impractical for real-time pressure measurements [35,36].

In addition to the commercially available compression therapy monitoring solutions, several research studies on pressure sensors and systems have been reported in the literature. Raj et al. [37] used water-filled polymerizing vinyl chloride (PVC) envelopes connected to an electrical pressure transducer to measure the interface pressure at four positions and reported that only within 6–8 h of daily routine applied pressure falls significantly. Hafner et al. [38] reported a silicone oil-filled pressure sensing system to train healthcare staff for an optimal compression in venous ulcer patient management; however, no details about the effect of temperature, hysteresis, and dynamic pressure are reported. Barbenel et al. [16] demonstrated a pressure sensing system for interface pressure using PVC probes filled with vegetable oil and was only limited to a pressure range of 0–37.5 mmHg. Burke et al. [4] developed an interface pressure monitoring system using four commercially available force sensors after integrated with a microcontroller and was capable to work in a range of 0–96 mmHg. However, there was observed a large hysteresis and lack of repeatability. Mehmood et al. [39] reported a telemetric mobile-based sub-bandage for monitoring the pressure and moisture of wounds but because of improper integration of commercial sensors, the system size was big. Casey et al. [8] reported a wearable capacitive flexible pressure sensing technology for sub-acute compression therapy monitoring. This flexible sensor array is built on active capacitor-based pressure sensing. Therefore, it requires a connected power supply and control unit. Farooqui et al. [40] reported a low-cost inkjet-printed wireless sensing system for chronic wound monitoring by measuring the pH level and physical pressure at the wound site. Rahimi et al. [41] has proposed an LC wireless strain sensor for wound monitoring by directly printing the conductive traces on the wound dressing but linearity was limited to 35% strain and no details reported about the repeatability and reliability of the system. Deng et al. [42] fabricated an LC wireless sensor for wound monitoring with a sensitivity of 270 kHz/mmHg in the range between 0 and 200 mmHg.

The majority of implantable or wearable sensors are based in LC systems due to the wireless communication between sensor and reader coil. Fonseca et al. [43] presented a very flexible wireless LC pressure sensor that was rollable and foldable to a compact shape for catheter-based delivery. This sensor was tested acutely in vivo for greater than 30 days in canine models simulating abdominal aortic aneurysms (AAA). Li et al. [44] reported a low power flexible sensor for intracranial pressure (ICP) monitoring, with a dual-mode operation in piezoelectric and capacitive modes, accuracy and reliability can be improved using dual-mode capability. Chen et al. [45] presented a wireless pressure sensor for continuous intraocular pressure monitoring of glaucoma patients with a long sensing distance and small physical form factor. Lei et al. [3] reported a flexible capacitive pressure sensor for plantar pressure measurements, different ratios of polydimethylsiloxane (PDMS) prepolymer and curing agent were mixed to improve the linearity by tweaking the stiffness factor.

The work presented here shows a flexible thin-film capacitive pressure sensor that can be fabricated using a simple and cost-effective etching process. The proposed sensor can be used in a wide range of medical applications, including intra and extracranial pressure, wound healing, and muscle rehabilitation monitoring; although, in this instance it has been designed mainly for interface pressure monitoring during compression therapy.

Considering varying ulcer sizes and lower limb curvatures, as well as different positions, two versions of the sensor with different sizes were fabricated, after optimization of their design parameters for best quality factor and resonance frequencies. Nevertheless, both sensors are LC resonant tank circuits and work on a capacitive sensing mechanism. The optimization of such parameters is reported as analytical results. In the experimental work, the performance of these sensors was evaluated over a pressure range of 0–100 mmHg. In addition, both sensors were also tested for a wider pressure range of 0–300 mmHg, as to suit a varying range of medical applications.

The rest of the paper is organized as follows: Section 2 describes the methodology, including the design, fabrication, and validation of the sensor; Section 3 presents the results obtained (analytical and experimental); Section 4 and Section 5 provide the final discussion and conclusions, respectively.

## 2. Materials and Methods

The proposed sensor is based on an LC resonance circuit, where the resonance frequency of the LC circuit is proportional to the applied pressure. The schematic diagram of the wireless sensing system is presented in Figure 1a. By placing multiple sensors under compression bandage as shown in Figure 1b, an array of wireless sensors can be formed to help in delivering a more controlled personalized compression therapy for the fast recovery of venous ulcers. A wearable readout band can keep records of pressure profiles during the daily routine.

### 2.1. Sensor Design

The LC sensor is designed as a disc capacitor made of two parallel circular plates, and the inductor is a planar circular spiral coil located around one of the capacitor electrodes suited for a flexible design for a bandage–skin interface. A geometrical representation of the sensor and reader are shown in Figure 2a,b respectively. The resonance frequency $\left(f_{o}\right)$ of the proposed LC sensor depends on the inductance $\left(L_{s}\right)$ and capacitance $\left(C_{s}\right)$ of the sensor, as given in Equation (1):(1)$$f_{o}=\frac{1}{2\pi \sqrt{L_{s}C_{s}}}$$

The capacitance of the sensor can be calculated as in Equation (2):(2)$$C_{s}=\frac{\epsilon_{o}\epsilon_{r}\pi r^{2}}{d}$$
where $\epsilon_{o}$ is the permittivity of free space, $\epsilon_{r}$ is the relative permittivity of dielectric material in the capacitor, and r is the radius of the disk capacitor. The inductance of the planar spiral inductor is calculated using its current sheet expression [46], which depends on the inner diameter $d_{in}$, outer diameter $d_{out}$ and number of turns N, as given in Equation (3):(3)$$L_{s}=\frac{\mu_{o}N^{2}d_{avg}C_{1}}{2}\left(\ln\left(C_{2}/\tau \right)+C_{3}\tau +C_{4}\tau^{2}\right)$$
where $\mu_{o}$ is the permeability of free space, N is the number of turns, $d_{avg}=\frac{(d_{in}+d_{out)}}{2}$, $\tau =\frac{\left(d_{out}-d_{in}\right)}{\left(d_{out}+d_{in}\right)}$, and $C_{1}$, $C_{2}$, $C_{3}$ and $C_{4}$ are the coefficients for the current sheet expression, which are 1, 2.46, 0, and 0.2 for a circular design [46].

#### Parasitic Components

The inductive part of the sensor, consisting of circular spirals, can be modeled accurately using lumped elements. Its elements are an inductor $\left(L_{s}\right)$, a parasitic resistance $\left(R_{tot}\right)$, and parasitic capacitance ($C_{p}$), where $L_{s}$ and $R_{tot}$ are in series in parallel to $C_{p}$ as shown in Figure 3a.

One of the major parasitic effects that play a major role in the quality factor of the inductor is the series resistance, which is modeled as $R_{tot}$ in this paper. A large $R_{tot}$ will result in a poor quality factor of the inductor in the sensor, as well as in the reader coil. This $R_{tot}$ can be represented by Equation (4), which includes direct current resistance $\left(R_{dc}\right)$ and alternating current resistance $\left(R_{ac}\right)$.
(4)$$R_{tot}=R_{dc}+R_{ac}$$

$R_{dc}$ can be calculated according to Equation (5), where ρ is the resistivity of the conductor, l is the length of the spiral conductor, w is the trace width and t is the trace thickness.
(5)$$R_{dc}=\frac{\rho l}{wt}$$

For a spiral inductor with N number of turns, outer and inner diameters $d_{out}$ and $d_{in}$, the length of the conductive traces can be calculated using Equation (6).
(6)$$l=\frac{\pi N\left(d_{in}+d_{out}\right)}{2}$$

The component $R_{ac}$ in Equation (4) is affected by the values of $R_{skin}$ and $R_{prox}$, which correspond to the skin effect and proximity effect, respectively:(7)$$R_{ac}=R_{skin}+R_{prox}$$

The skin effect occurs at higher frequencies when current does not flow through the complete cross-sectional area of the conductor, and it starts flowing only through its surface as shown in Figure 3b, which increases the effective resistance. In Figure 3b, the red color represents the skin depth (δ) for current flow and the blue color shows the area without electric current. This effect is represented by the skin depth δ. The mathematical expression to compute $R_{skin}$ is given in Equation (8) [47]. Here $\mu_{o}$ is the permeability constant and $\mu_{r}$ is the relative permeability of the conductor and f is the operational frequency.
(8)$$R_{skin}=\frac{\rho l}{w\delta \left(1-e^{\frac{-t}{\delta }}\right)\left(1+\frac{t}{w}\right)},\mathrm{where}\;\delta =\sqrt{\frac{\rho }{\pi \mu_{o}\mu_{r}f}}$$

The proximity effect is another major contributor to $R_{ac}$ that becomes significant above a frequency specific to the design, known as crowding frequency, $f_{crit}$. In the signal frequencies above $f_{crit}$, magnetic forces surrounding the conductor become significant and result in a nonuniform current flow through the conductor. This redistribution of the current causes an increase in effective resistance and can be calculated through Equation (9) [48].
(9)$$R_{prox}=\frac{R_{dc}}{10}{\left(\frac{f}{f_{crit}}\right)}^{2},\mathrm{where}\;f_{crit}=\frac{3.1\left(w+s\right)\rho }{2\pi \mu_{o}w^{2}t}$$

The parasitic capacitance between the nearby turns can be computed from Equation (10) [49,50], where α and β are 0.9 and 0.1, respectively, and represent the parasitic contribution due to the air gap between the coil turns, and the gap between the metallic tracks and the substrate, as shown in Figure 3c. $\epsilon_{rc}$ and $\epsilon_{rs}$ are the relative permittivity of air and substrate material respectively.
(10)$$C_{p}=C_{pc}+C_{ps}=\frac{lt\epsilon_{o}}{s}(\alpha \epsilon_{rc}+\beta \epsilon_{rs})$$

The value of the self-resonance frequency $f_{SRF}$ of an inductor is critical, as above this frequency the parasitic capacitance of the inductor becomes dominant. The $f_{SRF}$ can be calculated using Equation (11) [50].
(11)$$f_{SRF}=\frac{1}{2\pi \sqrt{L_{s}C_{p}}}$$

Finally, the quality factor of the LC sensor is given by Equation (12) [51].
(12)$$QF=\frac{1}{R_{tot}}\sqrt{\frac{L_{s}}{C_{s}}}$$

### 2.2. Device Fabrication

After the optimization of design parameters that is discussed in Section 3.1, a wet etching process was used to fabricate the two different sensors and their reader antennas. Figure 4 shows the stages in the fabrication process. In step I, as shown in Figure 4a, the mask of the sensor was directly printed on a 50 μm thick copper-coated polyimide film (Flexible isolating circuit 50 μm-coppered 35 μm-1 side, CIF, Buc, France) with a LaserJet printer (HP M553, HP Technology, Dublin, Ireland). In step II, the printed copper sheets were immersed in an etchant solution (CIF, Boosted ferric chloride solution). After manual stirring for 15 min at room temperature, all the unwanted copper was removed as shown in Figure 4b, and the patterned sheet was washed with hot water. Acetone was used to remove the ink particles from the copper surface after the etching process. In the next step, a polydimethylsiloxane (PDMS) layer (Ultra-thin film, 30° shore A hardness, Silex Ltd., Bordon, UK) of 200 μm thickness was cut into a circular shape equal to the diameter of the capacitor electrodes and was placed on the bottom electrode as shown in Figure 4c. PDMS is widely used as a dielectric layer in capacitive pressure sensors due to its low Young’s modulus and compressibility. An adhesive layer composed of polypropylene and synthetic rubber of 90 μm thickness (Tesa64621, Tesa, Norderstedt, Germany) was placed around the PDMS layer as shown in Figure 4d. In the final step, the top layer of the sensor was folded onto the PDMS layer for the final assembly of the sensor. Figure 4e,f shows the top and bottom views of the fabricated sensor. The reader antenna was also fabricated by the same etching procedure, and flexible multithread wires were soldered to connect with a Sub-Miniature version A (SMA) connector.

### 2.3. Device Validation

To test the fabricated system (sensor with reader coils), a bench-test model was developed using a vector network analyzer (VNA E5063, Keysight Technologies Inc., Santa Rosa, CA, USA), a high-pressure glass bottle (Pressure+ 1000, Duran, Mainz, Germany), and a digital pressure gauge (Traceable 3462, Fisher Scientific Ltd., Loughborough, UK), as shown in Figure 5. The sensor was placed inside the pressure chamber and its response recorded using the reader antenna, which was placed outside the wall of the chamber. The pressure was varied using a vacuum pump (FB70155 Pump, Fisher Scientific Ltd., Loughborough, UK) to produce positive pressure inside the chamber, which was measured as well by the digital pressure gauge. The input impedance of the VNA was 50 Ω. A frequency sweep was generated from the VNA to observe the variation in resonance frequency against the varying pressure, and the *S* parameters of the sensor were recorded simultaneously.

## 3. Results

The results presented in this paper comprise of the outcomes of two types of investigation: analytical investigations (Section 3.1) and experimental investigations (Section 3.2). The analytical investigations are performed for optimization of design parameters $\left(d_{out},N,w,s\right)$ to achieve the best quality factor (QF), and lower resonance frequencies $(f_{o}).$ The experimental investigations are performed to test and characterize the performance of the two fabricated prototype sensors on suitable testbeds.

### 3.1. Analytical Results: Numerical Estimation of Sensor Parameters

Sensor optimization was done in two steps. In the first step, the outer diameter $\left(d_{out}\right)$ and the number of turns (N) of the inductor were optimized while keeping the trace width (w) and trace separation (s) constant. In the second step, after selecting the optimal values of $d_{out}$ and N, both s and w were adjusted to achieve the best quality factor (QF) with a low resonance frequency $\left(f_{o}\right)$.

#### 3.1.1. Optimization of Outer Diameter $\left(d_{out}\right)$ and Number of Turns (N)

Before the fabrication stage of the sensor, MATLAB numerical modeling was performed to achieve the best quality factor (QF) within low resonance frequency $\left(f_{o}\right)$ range to achieve a better signal to noise ratio (SNR). The two different designs of the sensor, sensor 1 (S_1_) and sensor 2 (S_2_), were characterized according to their individual parameters. S_1_ was modeled for different $d_{out}$ values, between 36 and 45 mm, and a varying N from 1 to 10, while keeping s = w = 500 μm. As can be seen from the data point shown in Figure 6, the best QF was 106.4, with a correspondent resonance frequency of 17.147 MHz, when $d_{out}$ and N were 45 mm and 10 respectively. However, to keep the sensor size small, we selected $d_{out}$ = 40 mm and N = 10 for the fabrication as there was no significant loss in QF (97.46), and $f_{o}$ was also low (19.188 MHz).

A similar model was computed for S_2_ as shown in Figure 7. In this case, the objective was to design a relatively small sensor; therefore, $d_{out}$ was varied between 10 and 14 mm and N between 1 and 5 turns, while s and w were kept constant at 500 and 200 μm respectively. For S_2_, the highest QF was ~32, with a $f_{o}$ of 222.4 MHz for $d_{out}$ = 14 mm and N = 5; however, we selected $d_{out}$ = 12 mm and N = 5 to achieve an optimal set of QF (23.93) and $f_{o}$ (259.44 MHz) against the size of the sensor.

#### 3.1.2. Optimization of Trace width (w) and Trace separation (s)

Trace width and trace separation also affect the QF and resonance frequency; therefore, complete numerical modeling was performed for the selection of the trace geometry. QF and $f_{o}$ were analyzed for different values of s and w, while the number of turns and $d_{out}$ were fixed this time. Both trace width (w) and trace gap (s) were varied within the maximum allowable range to fit within the limits of given sensor size and number of turns. For S_1__,_ values of s and w were modeled between 200 and 600 μm and $d_{out}$ and N were 40 mm and 10, respectively. As shown in Figure 8, the highest QF (103.5) was observed for s = 325 μm and w = 400 μm, with a resonance frequency of 16.82 MHz. For an equally distributed pattern with a trace width (w) and trace gap (s) of 500 μm, a very small loss in QF (~5%) was observed, therefore, w = s = 500 μm were chosen for the design of S_1_.

S_2_ was modeled by varying s between 200 and 300 μm and w from 200 and 500 μm, while keeping $d_{out}$ = 12 mm and N = 5 fixed, as shown in Figure 9. Maximum QF was 23.93 with a resonance frequency of 259.44 MHz for a combination of s = 500 μm and w = 200 μm. As both the QF and the resonance frequency of S_2_ were very sensitive to trace width and separation, the combination of s and w that produced the best QF were chosen for S_2_.

### 3.2. Experimental Prototype and Results

After selecting the optimized design parameters $\left(d_{out},N,s,\mathrm{and}\;w\right)$, two sensors, of outer diameters 40 and 12 mm (shown in Figure 10), were fabricated and tested using the test-bench described in Section 2.3. The key design parameters, results, and operating frequencies for both sensors and respective reader coils are listed in Table 1.

As discussed in Section 1, since bandage pressure varies between 10 and 60 mmHg during compression therapy, both fabricated sensors were tested for a pressure range of 0 to 100 mmHg. The reader coil connected with the network analyzer was magnetically coupled with the sensor, and the response of the sensor over varying pressure was measured. The measurements from VNA were triggered at an interval of 5 mmHg for a narrow range of 0–100 mmHg. These measurements are the reflection coefficients (S11 parameter) and are shown in Figure 11 and Figure 12 for the sensors S_1_ and S_2_, respectively.

In addition to the compression therapy monitoring, the proposed sensors could be used for other medical applications, including physiological pressure measurement. Therefore, both sensors were also tested over a wider range of 0 to 300 mmHg that covers almost the entire physiological pressure range. The measurements from VNA were triggered at an interval of 25 mmHg for a wide range of 0–300 mmHg. Figure 13 and Figure 14 show the measured reflection coefficients (S11 parameter) of S_1_ and S_2_ over this broad pressure range.

As the response of both the sensors was linear within the targeted pressure range of 0 to 100 mmHg, a first-order polynomial was fitted over the measured response of the sensors. The coefficients of the linear fitted model are given in Table 2. The measure sensor response (dotted) and fitted curve (solid) for both sensors are shown in Figure 15.

In the sensor response over a wide range of pressure up to 300 mmHg, a nonlinearity, associated with compression saturation of the dielectric layer, was observed at higher pressures as shown in Figure 16. Therefore, a second-order polynomial function was fitted to the measured response to obtain a model relating the resonance frequency to the pressure. The values of R-square (goodness of fit) and the model coefficients are listed in Table 3.

To assess the repeatability of pressure measurement with both sensors, the response of the sensors for six different pressure points between 0 and 100 mmHg was measured repeatedly for 10 cycles. Figure 17 and Figure 18 show the repeatability of S_1_ and S_2_, respectively. The mean values of the frequency response against applied pressure $\left(f_{u}\right)$ and standard deviation (σ) of 10 repeated measurements at 6 pressure points (100, 80, 60, 40, 20, 0) mmHg are given in Table 4.

## 4. Discussion

An LC pressure sensing system is developed to measure the pressure in compression therapy due to wireless communication between sensor and reader coil. Optimization of the sensors is essential to achieve the best quality factor and resonance frequency while keeping the sensor size limited. Optimized values of outer diameter $\left(d_{out}\right)$ and the number of turns (N), trace width (w), and trace separation (s) are listed in Table 1. The parasitic components of the sensor which are parasitic capacitance and parasitic resistance at resonance frequency were analyzed through numerical modeling and their values are reported in Table 1. The reported sensors were fabricated using a wet etching process, which is cost-effective and very simple but comes at the cost of less control on trace widths. In these circumstances, the thinnest trace width achieved was 200 μm. Both sensors were characterized using a bench test setup that was developed during this research work. Both sensors showed good linearity and repeatability for a pressure <100 mmHg.

As shown in Figure 15, the response of both designed sensors was linear over a pressure range of 0–100 mmHg, with a sensitivity of 8 kHz/mmHg for S_1_ and 65 kHz/mmHg for S_2_. The sensor response was observed as nonlinear at the higher pressure range of 0 to 300 mmHg, as shown in Figure 16. This is due to the nonlinear effect of the compression saturation of the dielectric layer of the capacitor in the sensor. Up to 100 mmHg, the sensitivity of S_1_ was 8.11 kHz/mmHg, which was reduced at higher pressure due to the dielectric layer saturation. Similar behavior was noticed for S_2_, where sensitivity was 65.48 kHz/mmHg up to 100 mmHg, and was reduced when the sensor was loaded with higher values of pressure.

Both sensors offered good repeatability as shown in Figure 17 and Figure 18, for a pressure range <80 mmHg; however, variability in measurements started growing in the sensor response for higher applied pressures (>80 mmHg), due to the already mentioned hysteresis of the dielectric layer. As it can be noticed from Table 4 that the average repeatability for both the sensors over the pressure range of 0–100 mmHg is slightly larger than the sensitivity per mmHg, the measurement uncertainty is estimated as less than ±1 mmHg.

From Table 1, it can be noticed that QF of S_1_ was better than S_2_, which is due to the exponential increase of the ac resistance at higher frequencies for S_2_ caused by the skin effect. In addition, by comparing the amplitude of S parameters of both sensors in Figure 13 and Figure 14, it is quite clear that S_1_ has a better signal to noise ratio (SNR) compared to S_2_.

There was noticed a difference between the calculated and measured resonance frequencies of both sensors (S_1_ and S_2_) was due to numerous possible reasons. The first possible reason might be the value of the PDMS dielectric constant ($\epsilon_{r\_\mathrm{PDMS}})$, which reported between 2.3 and 2.8 in literature [52]; however, for this research $\epsilon_{r\_\mathrm{PDMS}}$ was selected 2.65 as stated in Table 1. The second reason for this difference might be due to the roughness of conductive traces caused by an over-etching effect during the fabrication process. This difference was greater for S_1_ due to the uneven distribution of the dielectric layer and air gaps between the capacitor plates, which were relatively bigger as compared to S_2_. In the future, a more controlled fabrication process can be used to improve the etching process and dielectric layer deposition to overcome the mismatch between analytical and real values of sensor parameters.

A comparison of the developed sensors with previously reported systems is given in Table 5. It includes sensors developed explicitly for wound compression therapy, and as an extension, implantable sensors that measure bodily pressures in different locations. Although not designed specifically for the application targeted in this work, these implantable sensors are based on the same sensing concept of LC systems and operate in similar pressure ranges (as shown in Table 5). From the observation of the values listed in the table, it is noticeable that the sensitivity of the S_2_ sensor, 65.48 kHz/mmHg, is comparable with the prototypes reported in the literature. This is, in the author’s view, a noteworthy achievement, considering the fact that the sensor proposed here is based on a very simple and non-expensive fabrication method. By contrast, most states of the art sensors are based on microfabrication techniques, which are very expensive and laborious.

## 5. Conclusions

This work presented the design of a wireless capacitive pressure sensor of low-cost fabrication for medical applications. In particular, the sensor is designed to be used for monitoring of compression therapy in venous leg ulcers. The sensor design was optimized to achieve an optimal quality factor and resonance frequency by numerical modeling of the design parameters. The proposed thin-film flexible wireless pressure sensor was fabricated using a simple and cost-effective fabrication method. Two versions of the sensors, with 40 and 12 mm outer diameters respectively, were developed and characterized between 0–100 and 0–300 mmHg to cover the pressure range of compression therapy and the nominal range of all other physiological applications. A bench test setup was also developed for sensor validation using a glass pressure bottle, pressure pump, and a network analyzer. Both sensors showed good sensitivity, linearity, and repeatability for the lower pressure regime (0–100 mmHg). A MATLAB curve-fitting tool was used to model the relationship between the shift in resonance frequency and the change in pressure.

The focus of this research work was on the early prototype development of the sensor, which is characterized by the benchtop model. However, in the future, improved and miniaturized prototypes will be fabricated by a more controlled fabrication process, and an extensive study will be performed on human subjects to validate the effectiveness. The miniaturization and replacement of the dielectric material used in the proposed sensors with other elastomeric polymers, can improve the linearity, sensitivity, and repeatability of the sensor and will make it more suitable for numerous medical applications.

## Figures and Tables

**Figure 1 sensors-20-06653-f001:**
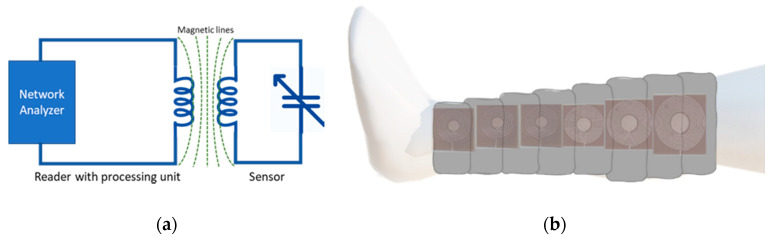
(**a**) Schematic diagram of wireless LC sensing system showing sensor and reader coil connected with a vector network analyzer (**b**) An application demonstration using flexible pressure sensors under the compression bandage.

**Figure 2 sensors-20-06653-f002:**
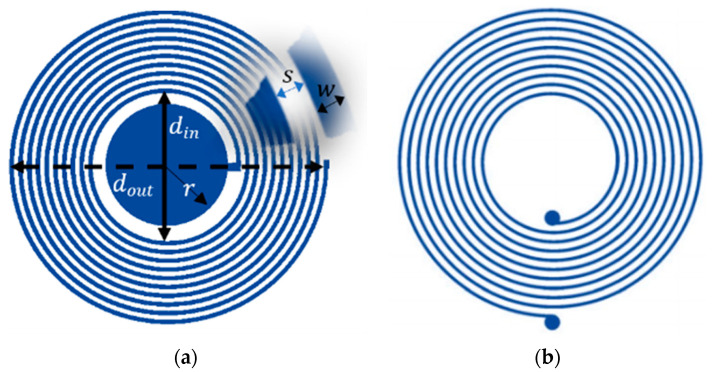
Geometrical representation of the proposed LC system: (**a**) LC sensor with a capacitor of the radius (r) and planar inductor with an inner diameter ($d_{in}$) shown with a solid line, outer diameter ($d_{out}$) shown with a dotted line, trace separation (s) and trace width (w); (**b**) Reader antenna with the same design parameters ($d_{in}$, $d_{out}$, s, w).

**Figure 3 sensors-20-06653-f003:**
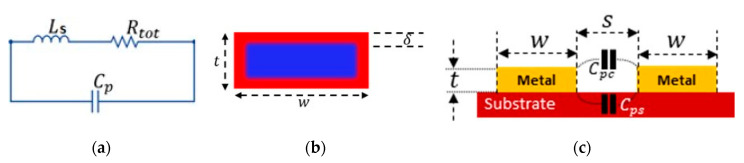
Parasitic effects: (**a**) Lumped model of the spiral inductor showing the inductor ($L_{s}$), parasitic resistance and capacitance; (**b**) Skin effect on a rectangular conductor, with current flowing only in the red area; (**c**) The parasitic capacitance is due to the air gap between coil turns $\left(C_{pc}\right)$ and the substrate material $\left(C_{ps}\right)$.

**Figure 4 sensors-20-06653-f004:**
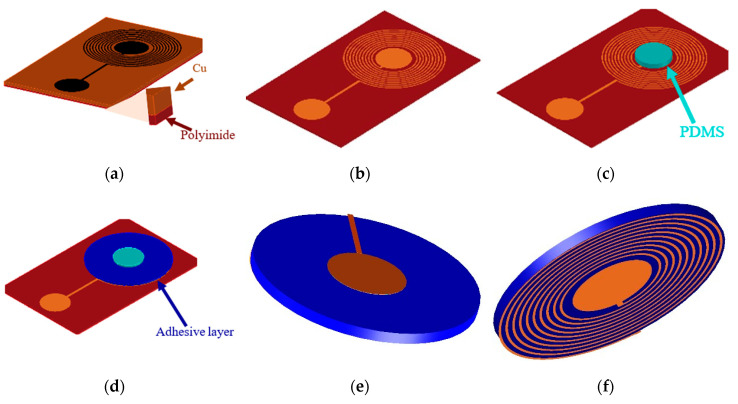
Fabrication process: (**a**) Copper-coated polyimide film with ink printed mask. (**b**) Etched pattern showing capacitor electrodes and planar inductor spirals. (**c**) Dielectric layer of PDMS elastomer. (**d**) Adhesive layer placement around the PDMS dielectric. (**e**) Top view and (**f**) bottom view of the LC sensor.

**Figure 5 sensors-20-06653-f005:**
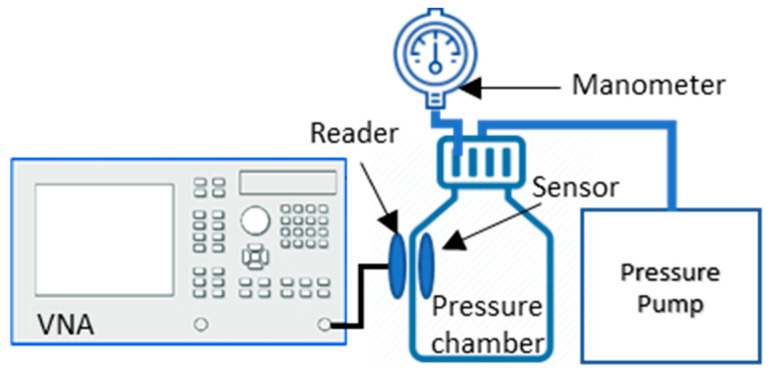
Bench test setup for sensor validation where reader coil is connected to vector network analyzer and sensor is kept inside the pressure chamber and pressure is varied using pressure pump.

**Figure 6 sensors-20-06653-f006:**
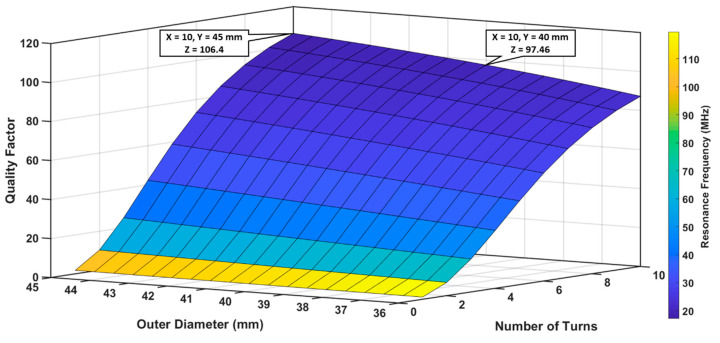
Analysis of S_1_ quality factor and resonance frequency for different number of turns and outer diameters, when trace separation and width were kept constant at 500 μm.

**Figure 7 sensors-20-06653-f007:**
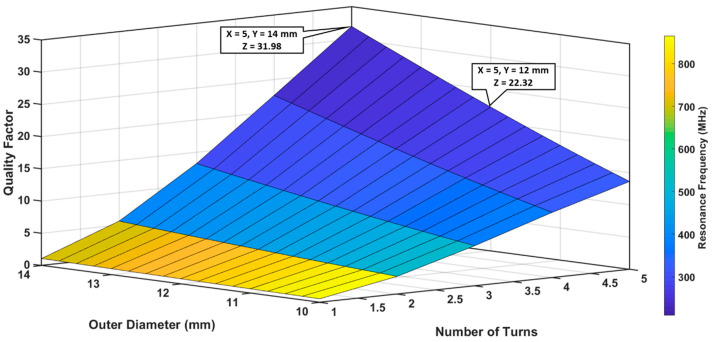
Analysis of S_2_ quality factor and resonance frequency for different number of turns and outer diameters, when trace separation was 500 μm and trace width was 200 μm.

**Figure 8 sensors-20-06653-f008:**
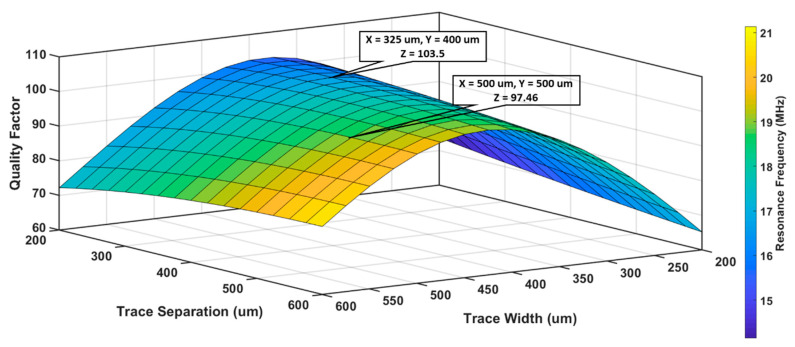
Analysis of S_1_ quality factor and resonance frequency for different trace separation and trace width, when the number of turns and outer diameter were 10 and 40 mm, respectively.

**Figure 9 sensors-20-06653-f009:**
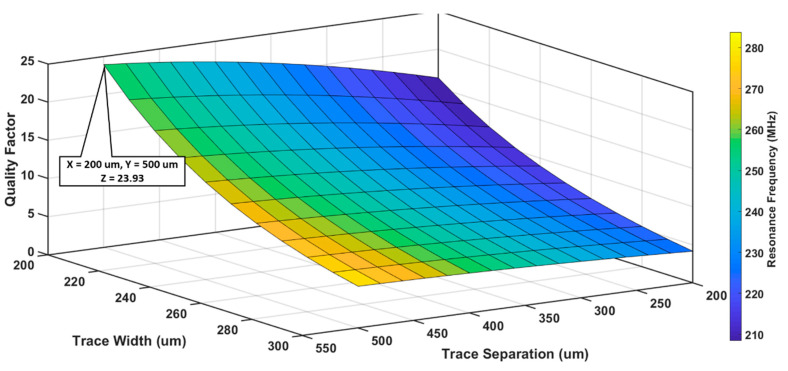
S_2_ quality factor and resonance frequency analysis for different trace separation and trace width when the number of turns and outer diameter were 5 and 12 mm, respectively.

**Figure 10 sensors-20-06653-f010:**
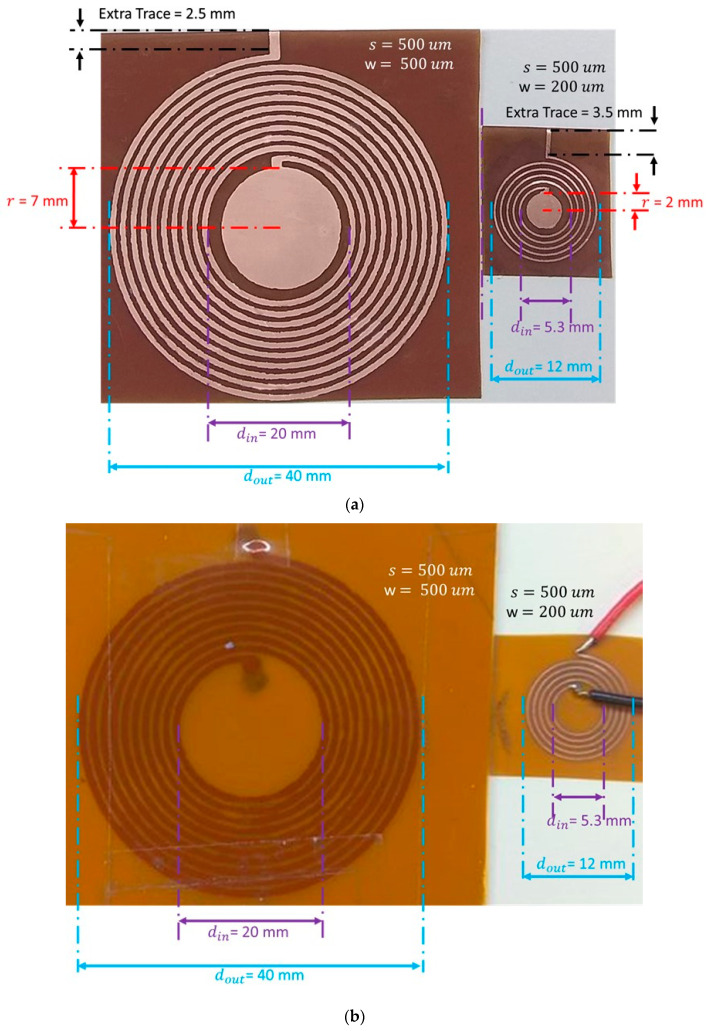
Fully labeled images of final prototypes, (**a**) Fabricated wireless LC resonance sensors: left, S_1_ (40 mm in diameter), and right, S_2_ (12 mm diameter); (**b**) Reader coils for S_1_ (**left**) and S_2_ (**right**).

**Figure 11 sensors-20-06653-f011:**
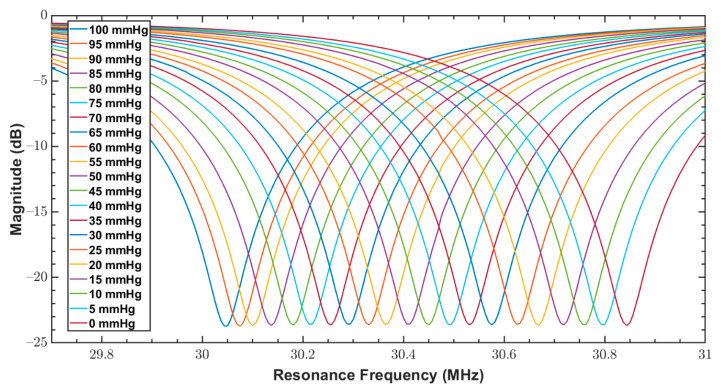
Reflection coefficients (S11 parameter) of S_1_ for a pressure range of 0 to 100 mmHg.

**Figure 12 sensors-20-06653-f012:**
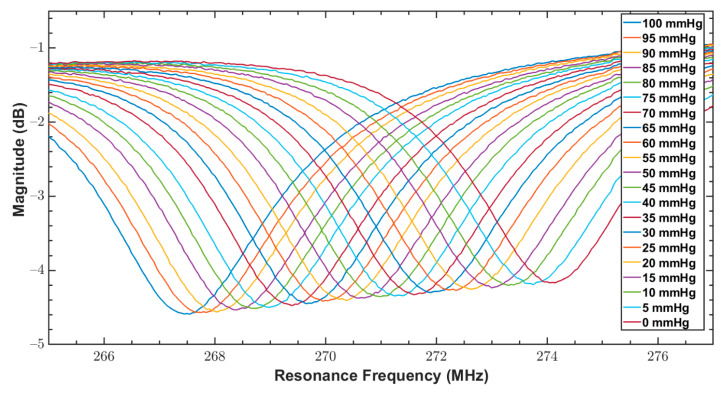
Reflection coefficients (S11 parameter) of S2 for a pressure range of 0 to 100 mmHg.

**Figure 13 sensors-20-06653-f013:**
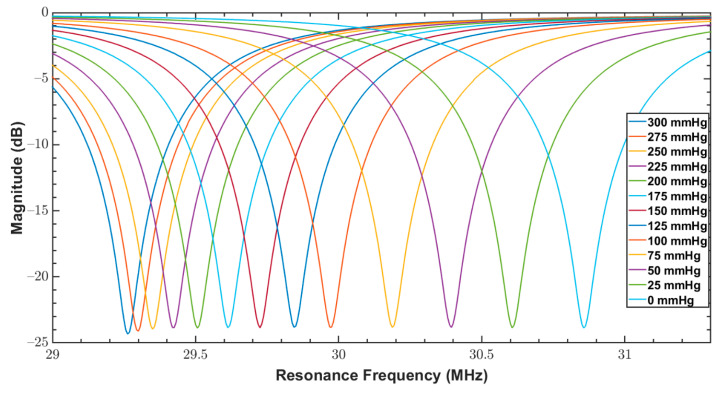
Reflection coefficients (S11 parameter) of S_1_ for a pressure range between 0 to 300 mmHg.

**Figure 14 sensors-20-06653-f014:**
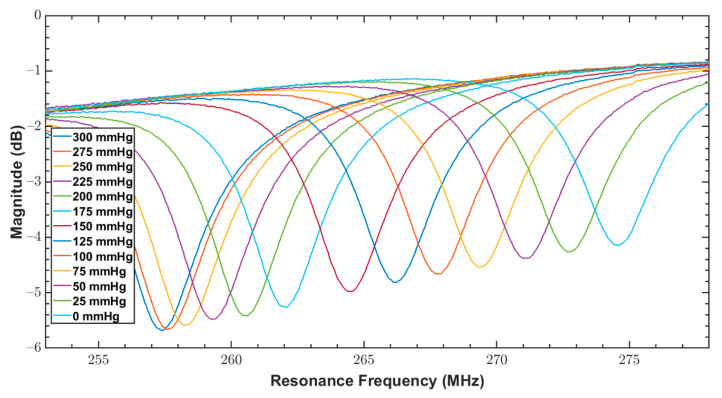
Reflection coefficients (S11 parameter) of S_2_ for a pressure range between 0 to 300 mmHg.

**Figure 15 sensors-20-06653-f015:**
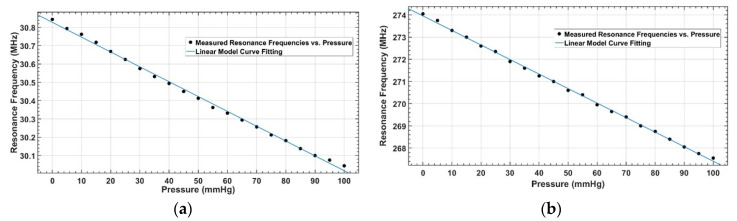
Linear model fitting for a pressure range of 0 to 100 mmHg: (**a**) Measured response (dotted) and linear fit (solid) of S_1_; (**b**) Measured response (dotted) and linear fit (solid) of S_2_.

**Figure 16 sensors-20-06653-f016:**
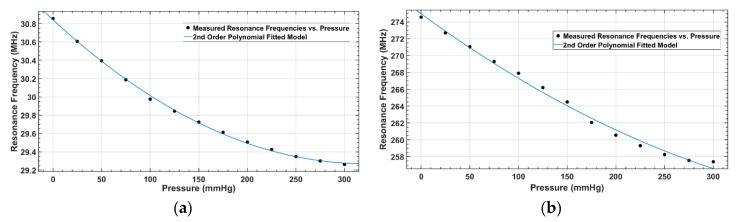
Nonlinear model fitting for a pressure range of 0 to 300 mmHg: (**a**) Measured response (dotted) and linear fit (solid) of S_1_; (**b**) Measured response (dotted) and linear fit (solid) of S_2_.

**Figure 17 sensors-20-06653-f017:**
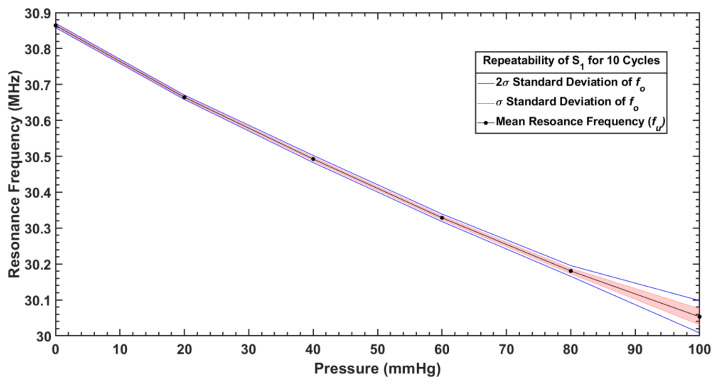
Repeatability of measurements with S_1_ over 10 cycles.

**Figure 18 sensors-20-06653-f018:**
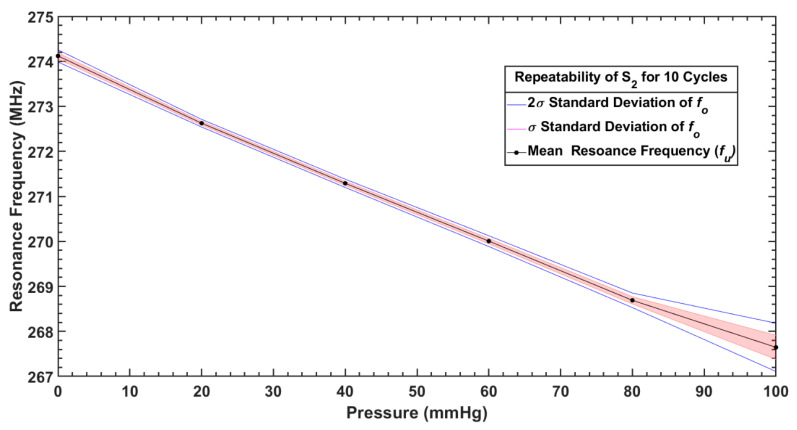
Repeatability of measurements with S_2_ over 10 cycles.

**Table 1 sensors-20-06653-t001:** Key design parameters and results for both sensors (S_1_, S_2_) and their readers (R_1_, R_2_).

Parameter/Result	S_1_/R_1_ (40 mm)	S_2_/R_2_ (12 mm)
Outer diameter, $d_{out}$ (mm)	40	12
Inner diameter, $d_{in}$ (mm)	20	5.3
Trace width, w (mm)	0.5	0.2
Trace separation, s (mm)	0.5	0.5
Number of turns, N	10	5
Capacitor electrode radius, r (mm)	7	2
Dielectric layer thickness, d (μm)	200	200
Spiral length, l (mm)	942.5	135.9
Skin depth, δ (μm)	15.15	4.12
DC resistance, $R_{dc}$ (Ω)	0.9371	0.3377
AC resistance, $R_{ac}$ (Ω)	3.7755	17.0536
Relative permittivity of PDMS [3], $\epsilon_{r\_\mathrm{PDMS}}$	2.65	2.65
Calculated resonance frequency, $f_{o\_\mathrm{cal}}$ (MHz)	19.188	259.44
Measured resonance frequency, $f_{o\_\mathrm{meas}}$ (MHz)	30.843750	274.10
Sensitivity, m (kHz/mmHg)	−8.110	−65.48
Capacitance, $C_{s}$ (pF)	18.06	1.4743
Parasitic capacitance, $C_{p}$ (pF)	0.72434	0.10442
Inductance, $L_{s}$ (μH)	3.8095	0.25572
Self-resonance frequency, $f_{SRF}$ (MHz)	95.81	974.80
Quality factor, QF at $f_{o}$	97.4584	23.92

**Table 2 sensors-20-06653-t002:** Coefficients of the polynomial equation (f(P)=m×P+β; where P is pressure) curve fitting between measured resonance frequencies and applied pressure.

Parameters	S_1_ (40 mm)	S_2_ (12 mm)
m (sensitivty)	−8.11 × 10^3^	−65.48 × 10^3^
β$(f_{o}$)	3.083 × 10^7^	2.74 × 10^8^
$R^{2}$ (goodness of fit)	0.9977	0.9989

**Table 3 sensors-20-06653-t003:** Coefficients of 2nd order polynomial ($f\left(P\right)=a\times P^{2}+b\times P+\beta$; where P is pressure) curve fitting between measured resonance frequencies and applied pressure.

Parameters	S_1_ (40 mm)	S_2_ (12 mm)
a	15.27	77.12
b	−9.797 × 10^3^	−8.439 × 10^4^
β$(f_{o}$)	3.084 × 10^7^	2.75 × 10^8^
$R^{2}$ (goodness of fit)	0.9991	0.993

**Table 4 sensors-20-06653-t004:** Mean value and standard deviation of measured resonance frequencies when both sensors were tested under different pressures for 10 cycles.

Sensor	Parameter	Pressure (mmHg)					
		100	80	60	40	20	0
**S_1_**	Mean, $f_{u}$ (MHz)	30.05	30.18	30.32	30.49	30.66	30.86
	Standard deviation, σ (kHz)	22.48	7.48	5.27	4.93	3.01	3.01
**S_2_**	Mean, $f_{u}$ (MHz)	267.64	268.69	270.00	271.29	272.62	274.12
	Standard deviation, σ (kHz)	26.81	80.96	59.86	45.94	42.49	67.49

**Table 5 sensors-20-06653-t005:** Comparative analysis of this study with the literature.

Study	Sensing Mechanism	Methodology	Linear Operational Range	Sensitivity	Fabrication Cost	Application
This study	Capacitive	LaserJet printing, Copper etching, Sandwiching of PDMS layer	0–100 mmHg	8.11 kHz/mmHg 65.48 kHz/mmHg	Low	Interface pressure monitoring during compression therapy
Deng et al. [42]	Capacitive	Si wafer moulding, PDMS casting, Conductive printing, Packaging	0–200 mmHg	270 kHz/mmHg	High	Wound monitoring
Casey et al. [8]	Capacitive	Micro-machining, Si wafer moulding, Electrodes patterning, Component mounting	10–80 mmHg	N/A	High	Sub-bandage pressure measurements
Farooqui et al. [40]	Capacitive	Screen printing, Conductive printing, Component mounting, Packaging	5–100 mmHg	0.0523 pF/mmHg	Medium	Smart bandage for chronic wounds
Fonseca et al. [43]	Capacitive	Standard lithography, Wet-chemical etching, Laser-cutting	70–120 mmHg	5.76 kHz/mmHg	High	Implantable pressure sensing
Chen et al. [45]	Capacitive	Oxide patterning, Coating, and patterning, Metal deposition, Deep Si etching	0–100 mmHg	160 kHz/mmHg	High	Intraocular pressure monitoring
Li et al. [44]	Capacitive	Microfabrication	0–50 mmHg	0.419 kHz/mmHg	High	Intracranial blood pressure monitoring
Rahimi et al. [41]	Inductive	Corona treatments, Laser patterning of mask, Screen printing of electrodes, Temperature curing	0–35%	150 kHz% strain	High	Wound monitoring
Mehmood et al. [39]	Resistive (FSR)	Off the shelf sensors integration with electronics, Biocompatible coating	0–60 mmHg	N/A	Medium	Sub-bandage pressure and wound moisture
Burke et al. [4]	Resistive (FSR)	Off the shelf sensors integration with electronics	0–96 mmHg	31.27 mV/mmHg	Medium	Sub-bandage pressure during venous compression therapy

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
