# Peer review of "Thin-Film Flexible Wireless Pressure Sensor for Continuous Pressure Monitoring in Medical Applications"

_sensors, 2020, doi:10.3390/s20226653_

Round 1

Reviewer 1 Report

The paper titled “Thin-film flexible wireless pressure sensor for continuous pressure monitoring in medical applications” details a very interesting topic and is well structure.

However, there some mistakes that it is necessary to modify to improve the quality of this work.

  1. The authors gave a lot of reference and the state of art is very good. However, they talked about previous studies but in the paper does not appear any comparison between their prototypes and the previous one.
  2. Line 139 there is a dot additional.
  3. The legend of figure 3 is too long. Some explanation should be included in the main text.
  4. In section 2.3, when the authors described the bench test setup, the impedance value of the VNA is not mentioned. The authors should be indicated the input impedance of the reader.
  5. In section 3.1.1 the results showed in the figure 6 are given by a measure or a simulation? It is a little bit confuse due to in the previous section a bench test set up has been described but in section 3, the authors mentioned a MATLAB software. Maybe it is necessary to rewrite this part.
  6. The paper describes two types of sensors, but what is the main difference between them?
  7. Table 1, the letter type in the 5 line is different.
  8. In the legend of figures 11, 12, 13 and 14 should indicate S11 instead of S parameter.
  9. In the Discussion section, the authors claim that the sensors were fabricated using a low-cost material, however this material is not indicated.
  10. A diagram with the final dimensions of the sensors should be included.

Reviewer 2 Report

The manuscript presents the design and evaluation of a device for continuous pressure monitoring in medical applications.

The work presents the design consideration and characterization. The work is well structured and contains a long tutorial on the design and evaluation process. Overall good.

The sensor proposed is based on a coil coupling in which the sensor has a variable capacitance which affects the resonance frequency. The variation is shown being detectable and proportional to pressure changes.

The model and the tests carried out in a pressure chamber seem validate the effectiveness of the sensor in detecting pressure changes, however, the tests have not been performed on subjects. Furthermore, the device is presented as continuous pressure monitor, but the results presented e.g. in figures 17, 18 show 6 pressure points over 10 cycles. Why not presenting the continuous sampling for 1 or more cycles? This applies to other results as well. Clarification/graphs on this point would strengthen the work.

S parameter should be clarified, if related to reflection coefficient in Figures 11,12,13,14 should be better explained. Reflection coefficient is briefly mentioned and is not completely clear from the caption or text.

Round 2

Reviewer 1 Report

The authors have included all the suggestion that I did in my first review.